# Effect of Carbonization Temperature on Microstructures and Properties of Electrospun Tantalum Carbide/Carbon Fibers

**DOI:** 10.3390/molecules28083430

**Published:** 2023-04-13

**Authors:** Hongtao Guo, Xiaofan Ma, Qiqi Lv, Chunmei Zhang, Gaigai Duan

**Affiliations:** 1Jiangsu Co-Innovation Center of Efficient Processing and Utilization of Forest Resources, International Innovation Center for Forest Chemicals and Materials, College of Materials Science and Engineering, Nanjing Forestry University, Nanjing 210037, China; 2Institute of Materials Science and Devices, School of Materials Science and Engineering, Suzhou University of Science and Technology, Suzhou 215009, China

**Keywords:** electrospinning, electrical property, pyrolysis temperature, carbonization

## Abstract

Compared with traditional metal materials, carbon-based materials have the advantages of low density, high conductivity, good chemical stability, etc., and can be used as reliable alternative materials in various fields. Among them, the carbon fiber conductive network constructed by electrospinning technology has the advantages of high porosity, high specific surface area and rich heterogeneous interface. In order to further improve the conductivity and mechanical properties of pure carbon fiber films, tantalum carbide (TaC) nanoparticles were selected as conductive fillers. The crystallization degree, electrical and mechanical properties of electrospun TaC/C nanofibers at different temperatures were investigated. As the carbonization temperature increases, the crystallization degree and electrical conductivity of the sample also increases, while the growth trend of electrical conductivity is markedly slowed. The best mechanical properties of 12.39 MPa was achieved when the carbonization temperature was 1200 °C. Finally, through comprehensive analysis and comparison, it can be concluded that a carbonization temperature of 1200 °C is the optimum.

## 1. Introduction

Electrospinning is a feasible and extensive technique for preparing nanofibers [1,2,3]. Electrospinning technology and the engineering applications of electrospun nanofibers have made remarkable progress, which has brought great convenience to human daily life [4,5,6] in the following fields: Medical field: Electrospinning technology can be used to prepare medical fiber tissue scaffolds, drug release systems, etc., which play important roles in promoting wound healing and treating disease. Environmental protection field: Due to the ability of electrospinning technology to prepare fibers with high specific surface area and small diameter, it can be used to prepare high-performance filter materials, protective clothing, air purifiers, etc., which help to improve indoor and outdoor air quality and protect the environment. Energy field: Electrospinning technology can be used to prepare nano-composite materials with good conductivity and corrosion resistance, which can be applied in solar cells, lithium-ion batteries and other fields. Textile field: Electrospinning technology can be used to prepare high-performance fibers and their products, such as bulletproof vests, lightweight aviation materials, high-end home textiles, etc., which are characterized by their light weight, softness, comfort, and high strength, improving the quality of human life.

In general, there are four methods for preparing micronano-sized fibers: (1) self-assembly, (2) phase separation, (3) template synthesis, and (4) electrospinning [7,8,9,10,11,12,13,14,15]. Such fibers have shown great application potential in electromagnetic shielding interference [16], energy storage [17], wearable intelligent devices, and other fields, by virtue of their advantages such as simple device composition, easy-to-understand operation, controllable diameter (micron to nanometer scale), and high specific surface area [18,19,20,21,22]. Electrospinning technology plays an important role in promoting the transformation and upgrading of traditional spinning technology, material innovation and fiber material functionalization. Electrospun carbon nanofibers can be prepared by electrospinning their precursor polymer nanofibers and then carbonizing them under high temperature. Precursor polymers such as polyacrylonitrile (PAN), polyurethane (PU) or polyvinylidene difluoride (PVDF) can be dissolved in N, N-dimethylformamide (DMF) for electrospinning [23,24,25,26]. Among these polymers, PAN is a linear polymer with a large number of cyanide side chains on its macromolecular chain. PAN has become the main raw material for carbon fiber production, and carbon fiber yield of more than 90% mass fraction can be obtained after carbonization [27].

Adding conductive nano-fillers into carbon fiber can form conductive pathways in the fiber, which is able to effectively improve the conductivity of the material [28,29,30,31,32]. In addition, the mechanical strength of nanofibers can be effectively enhanced by adding conductive fillers. In recent years, metal carbides have attracted much attention because of their excellent physical and chemical properties. Adding conductive nanoparticles to nanofibers can form a conductive path, which mainly involves two aspects: the excellent conductivity of the conductive nanoparticles themselves, and the interaction between the conductive nanoparticles and the nanofibers. Firstly, conductive nanoparticles have high electron mobility and carrier concentration, leading to their good conductivity. When conductive nanoparticles are added to nanofibers, they form a series of conductive paths inside the fibers, thus forming a conductive path. Secondly, there is also a certain interaction between the conductive nanoparticles and the nanofibers. During preparation processes such as electrospinning, conductive nanoparticles can interact with polymer chains in the polymer solution, thereby becoming encapsulated inside the nanofibers. This encapsulation not only enhances the interaction between the conductive nanoparticles and nanofibers but also prevents the conductive nanoparticles from falling off or aggregating during use. Therefore, by adding conductive nanoparticles, a conductive path can be formed in the nanofibers, giving them good conductivity. This conductivity can be applied in fields such as electronic devices, sensors, and photovoltaics, with broad application prospects. Among them, tantalum carbide (TaC) possesses excellent conductivity and hardness, and can be embedded into a carbon nanofiber matrix as a conductive filler [33,34]. TaC/C composite fiber materials with excellent electrical conductivity can be obtained by adding TaC nanoparticles into carbon nanofibers. However, the evolution process of Ta phase materials at different temperatures, the crystallinity of TaC at different temperatures, and the optimal carbonization temperature have not been explored.

In a previous work, Zhou et al. reported on the preparation of electrospun TaC/C nanofibers and the investigations into mechanical flexibility and conductivity [35]. However, the reasons for selecting 1200 °C as the carbonization temperature were not explained. Moreover, the phase changes of electrospun TaC/C nanofibers at different temperatures have not yet been studied. Herein, TaC/C nanofibers at different carbonization temperatures were prepared by electrospinning, pre-oxidation and high temperature heat treatment (Figure 1a). By analyzing the SEM, TEM and XRD of the target products at different carbonization temperatures, we can determine whether TaC nanoparticles were successfully introduced. The optimum carbonization temperature was obtained by a comprehensive comparison of crystallinity analyses, electrical conductivity and mechanical properties.

## 2. Results and Discussion

### 2.1. Surface Morphology

SEM images (Figure 1b–h) can clearly observe the surface morphology of TaCl_5_/PAN fibers electrospun with 40 wt% TaCl_5_ addition, and the microstructure of carbon-based composite fibers at different carbonization temperatures. The fiber has no obvious fracture phenomenon, which indicates that there is no agglomeration of nanoparticles in the process of electrospinning, that the spinning condition is good, and that the continuous fiber can be prepared successfully. After carbonization, the carbon skeleton can be preserved intact, which can form a spatial conductive network and provide conductive paths for a large number of free electrons to transport. The reduction of fiber volume after high temperature carbonization is attributed to the dehydrogenation and cyclization of PAN fiber during preoxidation and carbonization. The large molecular weight benzene ring becomes carbon, and the diameter of the fiber shrinks accordingly while the mass decreases. In addition, there is obvious adhesion between the fibers after 1200 °C, which may have a certain impact on the mechanical properties of the fiber membrane. Moreover, the reason why the fibers obtained at 1300 °C and 1400 °C are thicker than those obtained at 1100 °C and 1200 °C is that non-carbon atoms are discharged, making carbon atoms more concentrated, then form a hexagonal carbon net plane, and eventually forming a chaotic graphite structure as the carbonization temperature increases.

### 2.2. Crystalline Phase Characterization

Briefly, during the process of carbonization of TaCl_5_/PAN nanofibers, the carbonizing of PAN nanofibers is preceded by the removal of non-carbon atoms. It changes from a ladder structure to a carbonized structure with declined N atoms as the temperature rises in a high temperature environment (Figure 2a). During the process of polyacrylonitrile carbonization, polymer molecules gradually lose non-carbon elements and form highly pure carbon fiber structures. This is a complex process of substance transformation, involving various chemical and physical processes such as molecular depolymerization, gas-phase reaction, and solid-phase reaction. At high temperatures, polyacrylonitrile molecules first undergo cracking and loss of some hydrogen atoms, forming linear or cyclic macromolecules with nitrogen functional groups. Then, in the continuing heating process, the functional groups undergo chemical reactions such as dehydrogenation, backbone rearrangement, and carbon chain expansion, ultimately transforming into carbon fiber structures. At the same time, as the temperature increases, the carbonization reaction produces a large amount of gas, including hydrogen, nitrogen, ammonia, carbon monoxide, etc., which may participate in the carbonization reaction under certain conditions, further promoting the formation of carbon fiber structures. Meanwhile, the formation of the main product TaC/C can be attributed from the carbothermic reduction reaction, as shown in the following:PAN → C + CO ↑(1)
Ta_2_O_5_/TaO + C/CO → TaC + CO_2_ ↑(2)

The XRD test can be used to analyze the phase of the prepared electrospun carbon-based composite fiber. As can be seen from XRD (Figure 2b), as the carbonization temperature increases, the prepared Ta base composition phase exhibits obvious changes. As shown in Figure 2b, the XRD patterns of samples including S-800 and S-900 show that there are typical diffraction peaks at 2θ = 22.9°, 28.3° and 36.7°, corresponding to (00 1), (0 11 0) and (1 11 1) crystal planes, respectively, which are consistent with the standard XRD patterns of Ta_2_O_5_ phase (PDF#79-1375). At the carbonization temperature of 1000 °C and 1100 °C, the XRD pattern of samples shows that there are typical diffraction peaks at 2θ = 35.1°, 40.8°, 59.0° and 70.6°, corresponding to (1 1 1), (2 0 0), (2 2 0) and (3 1 1) crystal planes, respectively. This is consistent with the standard XRD pattern of TaO (PDF#78-0724). The samples of S-1200, S-1300 and S-1400 show that the diffraction peaks at 2θ = 40.5°, 58.7°, 70.1°, 73.8° and 87.7° correspond to the planes of (200), (220), (311), (222) and (400), respectively. This is similar to the standard XRD pattern of TaC phase (JCPDS#77 0205) [35]. The results show that the Ta phase in the fiber changes from Ta_2_O_5_ to TaO, and then changes from TaO to TaC phase during heating at 800 °C to 1400 °C. These results demonstrate that TaC is successfully embedded in carbon nanofibers. The formation of TaC nanoparticles was also proven by the X-ray photoelectron spectroscopy (XPS) analysis. The XPS analysis could be used to further investigate the chemical composition and the chemical status of related elements. The full XPS spectra of TaC/C fabrics is displayed in Figure 2c, from which C 1 s, Ta 4f and O 1 s were observed. For the Ta 4f core-level spectra (Figure 2d), four obvious peaks were found at 23.8 eV, 25.6 eV, 26.3 eV and 28.1 eV, respectively, associated with the Ta 4f_7/2_ (TaC), Ta 4f_5/2_ (Ta-C) and Ta 4f_5/2_ (Ta-O), respectively. Additionally, in the C 1s spectrum (Figure 2e), there were three characteristic peaks at 283.2 eV, 284.7 eV and 286.0 eV, which represented Ta–C, C–C and C–O, respectively. In addition, the small amount of oxygen in XPS could be derived from the pre-oxidization of PAN and the residual tantalum oxides. The flexibility of the TaC/C non-woven composite fabrics was illustrated by a digital image as shown in Figure 2f. The TaC/C electrospun composite fabrics can be curved with different bending angles (0°, 60°, 90°, 180°), and the luminance of the small bulb was not affected, demonstrating its excellent flexibility, which has the potential for application in wearable conductive materials.

In addition, we obtained the size of TaC grains dispersed in carbon nanofibers at 1200 °C~1400 °C from TEM images. As can be seen from Figure 3a–f, TaC nanoparticles can be well dispersed in carbon nanofibers. It can be seen from Figure 3g–i that the size of TaC nanoparticles changes from 5.8 ± 2.6 nm to 3.4 ± 1.9 nm at the carbonization temperature of 1200 °C~1400 °C. The higher the carbonization temperature, the smaller the grain size achieved. In addition, Jade software was used to calculate the degree of TaC grain crystallization at different carbonization temperatures.

According to Table 1, the crystallinity of TaC nanoparticles in carbon nanofibers varies with temperature. As the carbonization temperature increases, the crystallinity of different crystal planes of TaC shows an increasing trend, which indicates the better the crystal degree that can be obtained. However, better crystallinity does not necessarily mean better other properties. In the next section, we focus on the analysis of the electrical and mechanical properties of composite fiber membrane at different temperatures, and make further comparisons to obtain the optimal carbonization temperature.

### 2.3. Electrical and Mechanical Properties Analysis

By analyzing the electrical and mechanical properties of electrospun TaC/C fiber membranes at different carbonization temperatures, it can be concluded that carbonization temperature that has the optimal utilization value is the best carbonization temperature.

The electrical conductivity of electrospun TaC/C fabrics derives from the migration of free ions inside the conductive network as well as from the transport and hopping of electrons containing TaC nanoparticles. As can be seen from Figure 4a, with the increase of carbonization temperature, the electrical conductivity of TaC/C composite fiber gradually increases, and the electrical conductivity of sample S-1400 can reach 25.64 S·cm^−1^. The conductivity of S-900 to S-1200 samples shows obvious change, which is attributed to the transformation of Ta_2_O_5_ into TaC nanocrystals. The growth rate of the conductivity of the sample is obviously slowed down, which may be due to the fact that with the increase in carbonization temperature, the grain becomes finer. The continuity of the conductive network inside the fiber gradually deteriorates. However, thanks to the further increase in crystallinity and graphitization degree of TaC and carbon fiber, the conductivity of the sample does not show a downward trend.

The mechanical hysteresis effects of electrospun TaC/C fabrics were investigated using cyclic stretching–releasing tests under the applied strains of 2 mm/min. Figure 4b shows the mechanical strength of different samples. With the increase of carbonization temperature, the mechanical properties of the TaC/C fiber membrane are gradually enhanced, and the mechanical strength of S-1200 is the best, up to 12.39 MPa. However, the mechanical properties of S-1300 and S-1400 decreased, and their tensile strengths were 8.66 MPa and 7.07 MPa, respectively. In the process of carbonization, the tensile strength of the carbon fiber gradually increases, and reaches the maximum value when the packing density of the chaotic graphite structure keeps balance with the radial direction of the fiber. When the high carbonization temperature reaches the appropriate value, N element, namely non-carbon atoms, will discharge through the form of N_2_ with the continuous rise in temperature. Polyacrylonitrile is a nitrogen-containing high molecular weight compound, in which nitrogen atoms are covalently bonded to carbon atoms. During carbonization, as the temperature increases, the polyacrylonitrile molecules undergo pyrolysis reactions to produce gas and release solid residue (i.e., carbon fiber). In this process, nitrogen atoms are released from the molecule in the form of N_2_ because the breaking energy of N-N bonds is relatively small and N_2_ is highly stable [36]. Thus, the fiber porosity and defects will increase, resulting in a decrease in tensile strength. PAN molecules undergo a carbonization reaction at high temperatures, forming denser carbon fiber structures after losing non-carbon elements. However, at excessively high temperatures, the rate of depolymerization of PAN molecules increases, leading to too rapid removal of elements such as hydrogen, oxygen, and nitrogen, resulting in the formation of pores or defects in some parts of the fiber where atoms have not been fully converted into carbon. In addition, higher temperatures may cause changes in the crystal structure, further promoting the formation of pores and defects. These pores and defects weaken the mechanical properties of the carbon fiber, leading to a decrease in tensile strength. Through the above analysis and comprehensive comparison, 1200 °C is the best carbonization temperature.

## 3. Materials and Methods

### 3.1. Materials

Polyacrylonitrile (PAN, average M_w_ = 150,000) and Tantalum pentachloride (TaCl_5_, 99.9%) were supplied by J&K Scientific Ltd., Co., Ltd. (Beijing, China). N,N-Dimethylformamide (DMF, AR, 99.5%) was provided by Shanghai Aladdin Agent Co., Ltd. (Shanghai, China). All reagents were used without further purification.

### 3.2. Preparation of Electrospun TaC/C Composite Fabric

Preparation of TaCl_5_/PAN electrospinning precursor solution: a PAN solution with a mass fraction of 12 wt% was prepared by adding 1 g of PAN powder to 7.3 g of DMF solution. After stirring, 0.4 g of TaCl_5_ powder was added to the PAN solution, and the mixture solution of TaCl_5_/PAN was stirred on the magnetic stirrer for 12 h. In this study, in the process of pre-electrospinning pure PAN solution, we found that when the mass fraction of PAN was 12 wt%, continuous electrospinning could be achieved. The obtained fibers were uniform and had no obvious bead shape, indicating that nanofibers can be successfully prepared at this concentration. Then, the TaCl_5_/PAN mixture was transferred into a syringe (10 mL, inner diameter: 14.48 mm) with a stainless-steel needle. The static voltage used for electrospinning was 24~25 kV, and the thrust velocity of the syringe was 1.5 mL·h^−1^. TaCl_5_/PAN nanofibers were collected through aluminum foil on a drum (1500 rpm) at a distance of 15 cm between the needle and the collector. The ambient temperature was 25 °C and the relative humidity was 50% ± 5%. The resulting fiber membrane was placed in a vacuum oven at 80 °C for 4 h to dry. The dry fiber membrane was pressed with graphite plate and put into a high temperature tube furnace. Temperature program setting of tube furnace: temperature rise at 8 °C·min^−1^, temperature rise to 800 °C holding for 2 h, and this sample is recorded as S-800. Other samples with different carbonization temperatures were first held at 800 °C for 1 h, and then heated to 900 °C, 1000 °C, 1100 °C, 1200 °C, 1300 °C, and 1400 °C for 1 h at the rate of 8 °C·min^−1^. The samples were denoted as S-900, S-1000, S-1100, S-1200, S-1300, and S-1400.

### 3.3. Characterization

The morphology of the sample was observed by a field emission scanning electron microscopy system (SEM, JSM-7600F, JEOL Japan Electronics Co., Ltd., Tokyo, Japan) and transmission electron microscopy (TEM, JEM-2100 UHR, JEOL Japan Electronics Co., Ltd., Tokyo, Japan). The composition of the sample was analyzed by an X-ray diffraction analyzer (XRD, Ultima IV, Rigaku, Tokyo, Japan) from 2θ = 10° to 90° with 10°/min. X-ray photoelectron spectroscopy (XPS, AXIS Ultra, Urbana, IL, USA) was performed using an Al Kα radiation at a power of 225 W. The electrical conductivity of samples was measured by a four-point probe resistivity measurement system (SDY-4, Guangzhou Four-Point-Probe Technology Co., Ltd., Guangzhou, China). Four-point probe conductivity testing is a commonly used method for testing the electrical properties of materials, mainly used to measure the conductivity of solid, liquid, and semi-solid samples. Its testing principle is to form a four-corner measurement circuit in the sample through four independent electrodes, thereby accurately measuring the resistance and conductivity of the sample. The specific testing process is as follows: Prepare the sample to be tested and cut it into shapes and sizes (3 cm × 3 cm) that meet the standard requirements. Insert four probes into the sample and place them in specified positions. Two of the probes act as current probes, and the other two act as voltage probes. Apply a constant current or voltage to the sample, usually using an AC power supply. Measure the voltage and current values, and calculate the sample’s resistance. Calculate the sample’s conductivity based on the measured resistance value and the sample’s size. Note that when conducting a four-point probe test, the appropriate current or voltage intensity should be selected according to the characteristics of the sample, and strict compliance with testing standards and regulations is required. In addition, to ensure the accuracy and reproducibility of the test results, the precision and stability of the testing instrument should be strictly controlled, and external interference such as electromagnetic interference and temperature changes should be avoided during the testing process. Tensile tests of the electrospun composite fabrics were performed on a 3365 universal testing machine (Instron Test Equipment Trading Co., Ltd., Norwood, MA, USA) at a speed of 2 mm min^−1^. A universal material testing machine is a commonly used testing equipment mainly used for the mechanical performance testing of materials. Its testing principle is based on Newton’s third law and Hooke’s law, that is, when a material is subjected to an external force, a corresponding deformation and stress will occur, and the testing machine evaluates the strength, stiffness, toughness, and other mechanical properties of the sample by applying an external load and measuring the corresponding deformation and stress. The specific testing process is as follows: Prepare the sample to be tested and determine its geometric shape and size. Clamp the sample onto the testing machine and set the required test parameters, such as load size and displacement rate. Control the testing machine’s electric device to apply external loads, causing the sample to deform and respond with stress. At the same time, measure and record the sample’s indicators such as displacement, strain, and stress through sensors and other devices. Calculate the mechanical properties of the sample, such as modulus of elasticity, yield strength, elongation, etc., based on the acquired test data. Note that to ensure the accuracy and reliability of the test results, it is necessary to strictly control the testing environment and experimental conditions throughout the testing process and to comply with relevant testing standards and specifications. At the same time, regular calibration and maintenance of the testing equipment are required to ensure their stability and accuracy.

## 4. Conclusions

In this work, by introducing TaCl_5_ into PAN solution, one-dimensional electrospun TaC/C nanofibers were successfully prepared through the process of electrospinning, pre-oxidation and high temperature carbonization. In order to explore the best carbonization temperature, the best carbonization temperature was obtained through the characterization of SEM, TEM, XRD, electrical conductivity and mechanical properties, which lays a foundation for the multifunctional application of TaC/C fiber. As can be seen from SEM, TEM morphology characterization and XRD results, we successfully embedded TaC nanoparticles into carbon nanofibers to prepare a multi-scale carbon fiber-based spatial network, which provides the possibility for mobile carrier transport. The TaC nanoparticles are well embedded in the carbon nanofiber matrix, which further enhances the electrical conductivity. From the calculation and analysis of the crystallinity of TaC/C composite nanofibers at different temperatures, it is concluded that the grain size of TaC nanoparticles decreases with an increase in carbonization temperature. Through electrical conductivity and mechanical properties, the carbonization temperature has a significant impact on the various properties of TaC/C fiber.

## Figures and Tables

**Figure 1 molecules-28-03430-f001:**
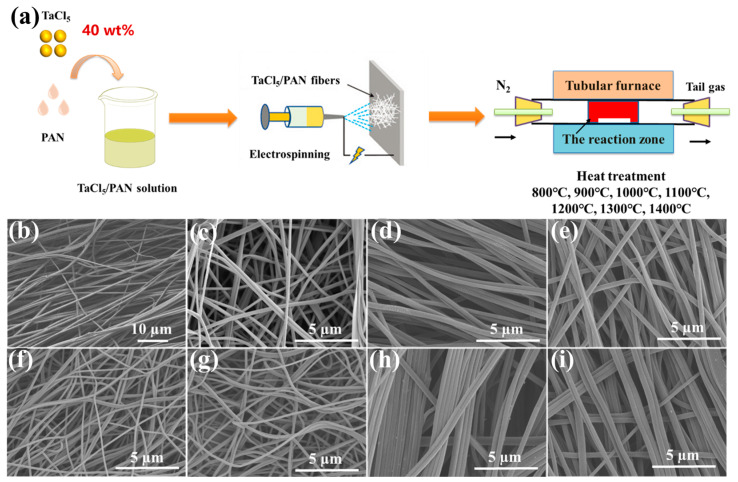
(**a**) Schematic diagram of preparation of electrospun TaC/C composite fibers at different carbonization temperatures. (**b**) SEM images of electrospun TaCl_5_/PAN fibers with 40 wt% TaCl_5_. SEM images of TaC/C composite fibers were prepared at different carbonization temperatures (**c**) 800 °C; (**d**) 900 °C; (**e**) 1000 °C; (**f**) 1100 °C; (**g**) 1200 °C; (**h**) 1300 °C and (**i**) 1400 °C.

**Figure 2 molecules-28-03430-f002:**
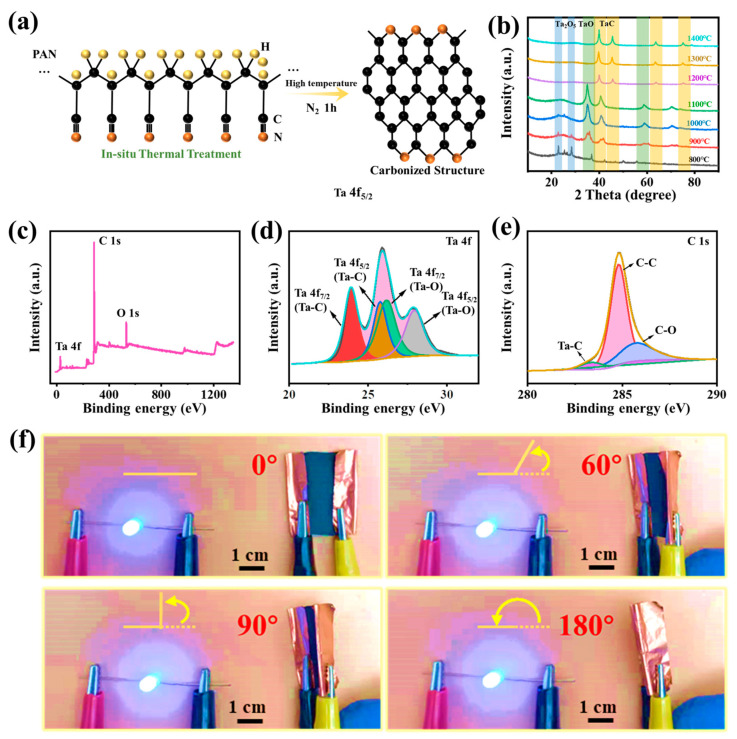
(**a**) The structural evolution of PAN under thermal treatment from the ladder structure to the carbonized structure. (**b**) XRD pattern of various samples at different carbonization temperatures. XPS survey scan of electrospun TaC/C fibers: (**c**) Full energy spectrum; (**d**) Ta 4f; (**e**) C 1 s. (**f**) Flexibility test of TaC/C electrospun fabrics at different bending angles.

**Figure 3 molecules-28-03430-f003:**
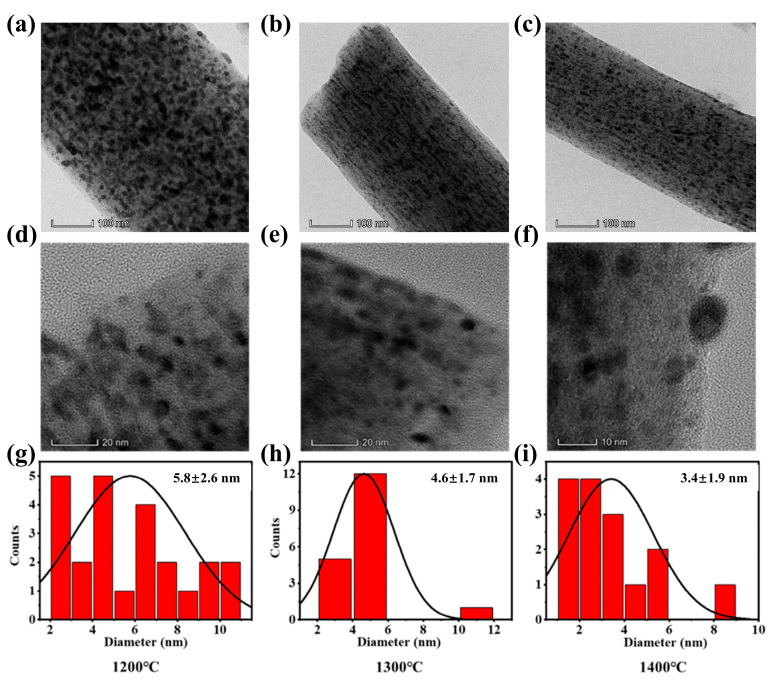
TEM images and grain size distributions of samples at different carbonization temperatures: (**a**,**d**,**g**) S-1200 (**b**,**e**,**h**) S-1300 and (**c**,**f**,**i**) S-1400.

**Figure 4 molecules-28-03430-f004:**
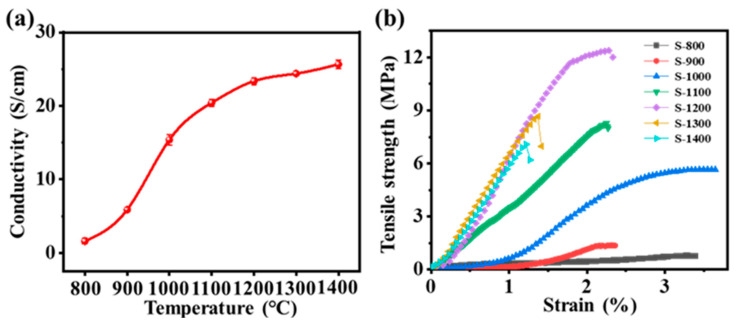
(**a**) Electrical conductivity and (**b**) tensile strength of electrospun TaC/C composite fabric samples at different carbonization temperatures.

**Table 1 molecules-28-03430-t001:** Crystallinity of TaC nanoparticles at carbonization temperatures from 1200 °C to 1400 °C.

Temperature (°C)	(1 1 1)	(2 0 0)	(2 2 0)	(3 1 1)	(2 2 2)
1200	86.45% ± 1.87%	92.51% ± 2.83%	96.69% ± 5.22%	97.18% ± 3.25%	99.30% ± 4.38%
1300	86.45% ± 1.87%	93.28% ± 3.99%	95.50% ± 5.26%	97.08% ± 3.59%	99.22% ± 2.34%
1400	95.81% ± 3.25%	97.78% ± 2.35%	99.46% ± 4.56%	99.43% ± 4.27%	99.86% ± 3.39%

## Data Availability

Data available from the corresponding author.

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
