# Peer review of "Effect of Carbonization Temperature on Microstructures and Properties of Electrospun Tantalum Carbide/Carbon Fibers"

_molecules, 2023, doi:10.3390/molecules28083430_

Round 1
Reviewer 1 Report (Previous Reviewer 1)
With the added experimental results, explanations, and modifications, I now approve this manuscript in its current form for publication in Molecules.
Author Response
Thank you very much for your recognition of our work.
Reviewer 2 Report (New Reviewer)
Comment 1: To demonstrate the chaotic graphite structure with the carbonization temperature increasing, the Raman characterizations of TaC/C samples at 800 ℃-1400 ℃ temperature should be measured.
Comment 2: The (HR)TEM characterizations of TaC/C samples at 1200 ℃-1400 ℃ should be added.
Comment 3: On Page 6 of 20, “when the high temperature carbonization temperature reaches the appropriate value N element, namely non-carbon atoms, will discharge through the form of N2 with the continuous rise of temperature”. How to prove this.
Comment 4: On Page 6 of 22, it said that “fiber porosity and defects will increase, resulting in a decrease in tensile strength”. It is recommended to use nitrogen adsorption/desorption measurements to confirm this.
Comment 5: The experimental part should be further described in detail, especially, the test process of the electrical and mechanical properties analysis.
Comment 6: Some errors need to be fixed. For example, in line 20 of paragraph 6, the reference 26 is not set to the same position as the others.
Comment 7: Some related papers (such as doi.org/10.1007/s12598-015-0586-2; doi.org/10.1007/s12598-022-02058-2; doi.org/10.1007/s12598-022-02146-3; doi.org/10.1021/acsenergylett.2c02203) are recommended to the authors for comparison and discussion.
Author Response
Response to the Comments (molecules-2309651)
Dear Editors and Reviewers:
Thank you for your letter and for the reviewers’ comments concerning our manuscript entitled “Effect of carbonization temperature on microstructures and properties of electrospun tantalum carbide/carbon fibers” (molecules-2309651). We have carefully studied the comments of the reviewers and revised the manuscript accordingly. These corrections are highlighted with red color while the typos and grammar issues are highlighted with blue color. A point-to-point response has been provided as following.
Reviewer:
- To demonstrate the chaotic graphite structure with the carbonization temperature increasing, the Raman characterizations of TaC/C samples at 800 ℃-1400 ℃ temperature should be measured.
Response: Thank you very much for your valuable comments. In this work, XRD test can demonstrate that the TaC/C nanoparticles are fabricated successfully when the carbonization temperature reaches 1200 ℃. And we can conclude that crystalline degree and graphitization degree exhibit an increasing trend from 1200 ℃ to 1400 ℃ through TEM images and calculation of crystallinity.
- The (HR)TEM characterizations of TaC/C samples at 1200 ℃-1400 ℃ should be added.
Response: Thanks very much for your suggestions. According to your comment, we have added the (HR)TEM characterizations of TaC/C samples at 1200 ℃-1400 ℃ in the revised manuscript.
- On Page 6 of 20, “when the high temperature carbonization temperature reaches the appropriate value N element, namely non-carbon atoms, will discharge through the form of N2 with the continuous rise of temperature”. How to prove this.
Response: Thanks a lot for your valuable comments. In the revised manuscript, we have explained the related reason why N element, namely non-carbon atoms, will discharge through the form of N2 with the continuous rise of temperature. The corresponding revisions are marked with red color.
“Polyacrylonitrile is a nitrogen-containing high molecular weight compound, in which nitrogen atoms are covalently bonded to carbon atoms. During carbonization, as the temperature increases, the polyacrylonitrile molecules undergo pyrolysis reactions to produce gas and release solid residue (i.e., carbon fiber). In this process, nitrogen atoms are released from the molecule in the form of N2 because the breaking energy of N-N bonds is relatively small and N2 is highly stable.”
- On Page 6 of 22, it said that “fiber porosity and defects will increase, resulting in a decrease in tensile strength”. It is recommended to use nitrogen adsorption/desorption measurements to confirm this.
Response: Thanks a lot for your comments. According to your comments, we have explained why fiber porosity and defects increase and tensile strength will decrease correspondingly. The corresponding revisions are marked with red color.
“PAN molecules undergo carbonization reaction at high temperatures, forming denser carbon fiber structures after losing non-carbon elements. However, at excessively high temperatures, the rate of depolymerization of PAN molecules increases, leading to too rapid removal of elements such as hydrogen, oxygen, and nitrogen, resulting in the formation of pores or defects in some parts of the fiber where atoms have not been fully converted into carbon. In addition, higher temperatures may cause changes in the crystal structure, further promoting the formation of pores and defects. These pores and defects weaken the mechanical properties of the carbon fiber, leading to a decrease in tensile strength.”
- The experimental part should be further described in detail, especially, the test process of the electrical and mechanical properties analysis.
Response: Thank you very much for your comments. According to your comments, we have further described the experimental part including the test process of the electrical and mechanical properties analysis in detail. The corresponding revisions are marked with red color.
“The electrical conductivity of electrospun TaC/C fabrics derives from the migration of free ions inside the conductive network as well as the transport and hopping of electrons containing TaC nanoparticles. The electrical conductivity of samples was measured by a four-probe method.”
“The mechanical hysteresis effects of electrospun TaC/C fabrics were investigated using cyclic stretching-releasing tests under the applied strains of 2 mm/min.”
- Some errors need to be fixed. For example, in line 20 of paragraph 6, the reference 26 is not set to the same position as the others.
Response: Thanks a lot for your comments. We have corrected related errors in the revised manuscript with blue color.
- Some related papers (such as doi.org/10.1007/s12598-015-0586-2; doi.org/10.1007/s12598-022-02058-2; doi.org/10.1007/s12598-022-02146-3; doi.org/10.1021/acsenergylett.2c02203) are recommended to the authors for comparison and discussion.
Response: Thank you very much for your comments. In the revised manuscript, we have cited related reference in corresponding place.
Round 2
Reviewer 2 Report (New Reviewer)
-
The revised manuscript has been greatly improved and is recommended for acceptance.
This manuscript is a resubmission of an earlier submission. The following is a list of the peer review reports and author responses from that submission.
Round 1
Reviewer 1 Report
The manuscript, “Effect of carbonization temperature on microstructures and properties of electrospun tantalum carbide/carbon fibers,” demonstrates the effect of carbonization temperature on the degree of crystallization, mechanical strength, and electrical conductivity of electrospun tantalum carbide/carbon nanofibers. To me, the work lacks in novelty and is not suitable for publication in its current form. Similar studies have already been published in “Materials Letters 200 (2017) 97-100” (Flexible and refractory tantalum carbide-carbon electrospun nanofibers with high modulus and electric conductivity). The authors have cited this work (Reference 16 in the manuscript) but have not discussed for the similarity. PAN electrospinning and then making carbon nanofibers has been reported numerous times in the literature, and tantalum carbide (TaC) is well known for its excellent conductivity and hardness and has been used as a conductive filler.
To improve the quality of the current work, the authors shall show some real applications, such as in flexible electronic devices, nanofiber-reinforced composites, and refractory materials using the synthesized product. Also, they shall improve the quality of the TEM images and perform more characterizations, such as bending tests and XPS.
Author Response
Response to the Comments (molecules-2150552)
Dear Editors and Reviewers:
Thank you for your letter and for the reviewers’ comments concerning our manuscript entitled “Effect of carbonization temperature on microstructures and properties of electrospun tantalum carbide/carbon fibers” (molecules-2150552). We have carefully studied the comments of the reviewers and revised the manuscript accordingly. These corrections are highlighted with red color.
Reviewer:
The manuscript, “Effect of carbonization temperature on microstructures and properties of electrospun tantalum carbide/carbon fibers,” demonstrates the effect of carbonization temperature on the degree of crystallization, mechanical strength, and electrical conductivity of electrospun tantalum carbide/carbon nanofibers. To me, the work lacks in novelty and is not suitable for publication in its current form. Similar studies have already been published in “Materials Letters 200 (2017) 97-100” (Flexible and refractory tantalum carbide-carbon electrospun nanofibers with high modulus and electric conductivity). The authors have cited this work (Reference 16 in the manuscript) but have not discussed for the similarity. PAN electrospinning and then making carbon nanofibers has been reported numerous times in the literature, and tantalum carbide (TaC) is well known for its excellent conductivity and hardness and has been used as a conductive filler.
Response: Thanks very much for your comments. Different from “Materials Letters 200 (2017) 97-100” (Flexible and refractory tantalum carbide-carbon electrospun nanofibers with high modulus and electric conductivity), we focus on the optimal carbonized temperature of electrospun TaC/C nanofibers and find that 1200 ℃ is the most suitable temperature through comparing crystallinity, electrical conductivity and tensile strength with various carbonized temperature.
To improve the quality of the current work, the authors shall show some real applications, such as in flexible electronic devices, nanofiber-reinforced composites, and refractory materials using the synthesized product. Also, they shall improve the quality of the TEM images and perform more characterizations, such as bending tests and XPS.
Response: Thanks very much for your comments. According to your suggestions, we have supplemented some real applications in the revised manuscript. Meanwhile, we improved the quality of the TEM images and added XPS analysis and bending demonstration. These corrections are highlighted with red color in the revision.
Figure 2. (a) The structural evolution of PAN under thermal treatment from the ladder structure to carbonized structure. (b) XRD pattern of various samples at different carbonization temperatures. XPS survey scan of electrospun TaC/C fibers: (c) Full energy spectrum; (d) C 1 s; (e) Ta 4f. (f) Flexibility test of TaC/C electrospun fabrics at different bending angles.
“The formation of TaC nanoparticles was also proven by the X-ray photoelectron spec-troscopy (XPS) analysis. The XPS analysis could further investigate chemical composi-tion and the chemical status of related elements. The full XPS spectra of TaC/C fabrics was displayed in Figure 2c, from which C 1 s, Ta 4f and and O 1 s were observed. For the Ta 4f core-level spectra (Figure 2d), three obvious peaks were found at 23.8 eV, 25.8 eV and 27.1 eV, and which were credited to Ta 4f7/2 (TaC), Ta 4f5/2 (Ta-C) and Ta 4f5/2 (Ta-O), respectively, thus demonstrating the presence of the carbonization state of Ta. Two main peaks located at 283.4 eV, 284.9 eV and 285.8 eV in XPS core-level spectra of C 1 s (Figure 2e), which can be assigned to Ta–C, C–C, respectively. The flexibility of the TaC/C non-woven composite fabrics was illustrated by displaying digital image as shown in Figure 2f. The flexibility of the TaC/C non-woven composite fabrics was illustrated by displaying digital image as shown in Figure 2f. The TaC/C electro-spun composite fabrics can be curved with different bending angles (0°, 60°, 90°, 180°), and the luminance of small bulb was not affected, which demonstrated its excellent flexibility, which has potential to be applied for wearable conductive materials.”
Reviewer 2 Report
The manuscript reported the effect of carbonization temperature on microstructures and properties of electrospun tantalum carbide/carbon fibers. The research topic is interesting. There are several points need be fixed before the possible publication.
(1) There are two “ (e) ” figures in Figure 1 and there is not “ (i) ” figure in the Figure.
(2) Were TaC/C composite fibers prepared at different carbonization temperatures from the same set of samples? Why were the fibers thicker obtained at 1300 oC and 1400 oC than that obtained at 1100 oC and 1200 oC?
(3) More references are needed in order to fully explain the significance and progress of the research.
Author Response
Response to the Comments (molecules-2150552)
Dear Editors and Reviewers:
Thank you for your letter and for the reviewers’ comments concerning our manuscript entitled “Effect of carbonization temperature on microstructures and properties of electrospun tantalum carbide/carbon fibers” (molecules-2150552). We have carefully studied the comments of the reviewers and revised the manuscript accordingly. These corrections are highlighted with red color while the typos and grammar issues are highlighted with blue color. A point-to-point response has been provided as following.
Reviewer:
The manuscript reported the effect of carbonization temperature on microstructures and properties of electrospun tantalum carbide/carbon fibers. The research topic is interesting. There are several points need be fixed before the possible publication.
- There are two “ (e) ” figures in Figure 1 and there is not “ (i) ” figure in the Figure.
Response: Thanks a lot for your comments. In the revised manuscript, we made corresponding corrections with red color and further checked the whole manuscript.
- (1) Were TaC/C composite fibers prepared at different carbonization temperatures from the same set of samples? (2) Why were the fibers thicker obtained at 1300 ℃ and 1400 ℃ than that obtained at 1100 ℃ and 1200 ℃?
Response: Thanks very much for your suggestions. The corresponding revisions are marked with red color.
- Yes, the TaC/C composite fibers were prepared at 900 ℃, 1000 ℃, 1100 ℃, 1200 ℃, 1300 ℃ and 1400 ℃ from the same set of samples.
- The reason why the fibers thicker obtained at 1300 ℃ and 1400 ℃ than that obtained at 1100 ℃ and 1200 ℃ is that non-carbon atoms will be discharged, making carbon atoms more concentrated, then forming hexagonal carbon net plane, and eventually forming chaotic graphite structure with the temperature carbonization temperature increasing. In this process, the tensile strength of carbon fiber gradually increases, and reaches the maximum value when the packing density of the chaotic graphite structure keeps balance with the radial direction of the fiber. When the high temperature carbonization temperature reaches the appropriate value N element, namely non-carbon atoms, will discharge through the form of N2 with the continuous rise of temperature. Thus, fiber porosity and defects will increase, resulting in a decrease in tensile strength, as shown in the following diagram.
“Moreover, the reason why the fibers thicker obtained at 1300 ℃ and 1400 ℃ than that obtained at 1100 ℃ and 1200 ℃ is that non-carbon atoms will be discharged, making carbon atoms more concentrated, then forming hexagonal carbon net plane, and eventually forming chaotic graphite structure with the temperature carbonization temperature increasing.”
- More references are needed in order to fully explain the significance and progress of the research.
Response: Thanks for your comments. According to your comments, we have cited more references in order to fully explain the significance and progress of the research.
Round 2
Reviewer 1 Report
The authors claimed that the novelty of the work lies in their study to optimize the carbonized temperature of electrospun TaC/C nanofibers (1200 ℃), which is the most suitable temperature through comparing crystallinity, electrical conductivity, and tensile strengths. However, the published paper (Flexible and refractory tantalum carbide-carbon electrospun nanofibers with high modulus and electric conductivity) already reported this temperature and has performed the study in more detail way. The authors have not discussed this paper in the introduction. Also, it is well-known that increasing the carbonization temperature up to ~1200 °C increases both the tensile strength and the modulus of CFs (https://www.nature.com/articles/srep22988, Strengthened PAN-based carbon fibers obtained by slow heating rate carbonization)
The authors showed the flexibility test of TaC/C electrospun fabrics at different bending angles, but it seems that the nanofiber part is not bent at all, and only the copper foil was folded. The test should be performed only using the nanofiber mat.
This paper lacks novelty, so I can not recommend it for publication. Instead, authors shall try any other suitable journal.
Some more points the authors should pay attention:
1. In the XPS survey scan of electrospun TaC/C fibers, where the “O” comes from?
2. In figure 2d, they should notate the peaks like this, Ta 4f7/2 (Ta-C), Ta 4f5/2 (Ta-C) and Ta 4f5/2 (Ta-O). Also, change the figure caption to 2d (Ta 4f) and 2e (C 1s).
3. Check the heading of section 3.2 Preparation of electrospun carbon composite fabrics with TaC/Fe3C-Fe nanoparticles. The authors should be more careful.
Reviewer 2 Report
The authors have revised the manuscript carefully according to the comments. The manuscript can be accepted in present form.